# Texture and semantic integrated small objects detection in foggy scenes

**Zhengyun Fang[1]☯, Hongbin Wang[2,3]☯\*, Shilin Li[4]‡, Yi Hu[2,3]‡, Xingbo Han[2,3]‡**

**1** College of Land Resource Engineering, Kunming Universityof Science and Technology, Kunming, Yunnan, China, **2** Faculty of Information Engineering and Automation, Kunming University of Science and Technology, Kunming, Yunnan, China, **3** Yunnan Key Laboratory of Artificial Intelligence, Kunming University of Science and Technology, Kunming, Yunnan, China, **4** Eleictric Power Reasearch Institute of Yunnan Power Grid Co., Ltd., Kunming, Yunnan, China

☯ These authors contributed equally to this work.
‡ SL, YH and XH also contributed equally to this work.
\* whbin2007@126.com

**Data Availability Statement:** A large number of experiments were carried out on two available large-scale data sets, i.e. "Cityscape to Foggy"(Hahner. M, Dai. D, Sakaridis. C, Zaech. J and Gool. L. V. Semantic Understanding of Foggy

## Abstract

In recent years, small objects detection has received extensive attention from scholars for its important value in application. Some effective methods for small objects detection have been proposed. However, the data collected in real scenes are often foggy images, so the models trained with these methods are difficult to extract discriminative object features from such images. In addition, the existing small objects detection algorithms ignore the texture information and high-level semantic information of tiny objects, which limits the improvement of detection performance. Aiming at the above problems, this paper proposes a texture and semantic integrated small objects detection in foggy scenes. The algorithm focuses on extracting discriminative features unaffected by the environment, and obtaining texture information and high-level semantic information of small objects. Specifically, considering the adverse impact of foggy images on recognition performance, a knowledge guidance module is designed, and the discriminative features extracted from clear images by the model are used to guide the network to learn foggy images. Second, the features of high-resolution images and low-resolution images are extracted, and the adversarial learning method is adopted to train the model to give the network the ability to obtain the texture information of tiny objects from low-resolution images. Finally, an attention mechanism is constructed between feature maps of the same scale and different scales to further enrich the high-level semantic information of small objects. A large number of experiments have been conducted on data sets such as "Cityscape to Foggy" and "CoCo". The mean prediction accuracy (mAP) has reached 46.2% on "Cityscape to Fogg", and 33.3% on "CoCo", which fully proves the effectiveness and superiority of the proposed method.

## Introduction

Object detection is one of the most important tasks in machine vision. It aims to find the object in images by computer and determine its category and position. With the advent of big

Scenes with Purely Synthetic Data[C]. IEEE Intelligent Transportation Systems Conference (ITSC), Auckland, New Zealand, 2019:3675-3681.) and "CoCo"(Lin. T, Maire. M, Belongie. S, Hays. J, Perona. P, Ramanan. D, Dollar. P and Zitnick. C. Microsoft COCO: Common Objects in Context[C]. European Conference on Computer Vision (ECCV), Zurich, Switzerland, 2014: 740-755.). The datasets can be downloaded from the below links: Cityscape to Foggy Datasets: https://people.ee.ethz.ch/∼csakarid/SFSU_synthetic/ CoCo Datasets https://cocodataset.org/#home.

**Funding:** The author(s) received no specific funding for this work.

**Competing interests:** The authors have declared that no competing interests exist.

data technology and deep learning in recent years, great breakthroughs have been made in object detection tasks [1–6]. However, the data collected in real scenes often contains small-sized objects. This kind of object contains very little information which may be lost after multi-layer convolution, so its discriminative features are difficult to extract, resulting in low model detection performance.

The existing small objects detection algorithms can effectively address the above problems. Such methods can be roughly divided into three categories: small objects detection methods based on multi-scale feature extraction [7–11], high-resolution feature-assisted small objects detection methods [12–16] and small objects detection methods guided by content information [17–21]. These methods focus on how to effectively improve the representational capability of the model to extract the discriminative features of small objects. Nevertheless, due to the influence of weather conditions, the data collected in real world are often foggy images. The quality of such images are often impared, which will cause the model's characterization ability to decline during small objects detection. This makes it difficult for the above methods to extract the discriminative object features. In addition, the existing small objects detection algorithms ignore the texture information and high-level semantic information of small objects, which partly limits the detection performance. As shown in Fig 1 below, the street view images collected in a foggy scene are low in sharpness and contrast, so the detection model trained directly on such images does not have the ability to extract the discriminant features of the object. It can also be seen from Fig 1 that it is difficult to obtain the texture information of the road indication sign in the low-resolution foggy images. Moreover, the road signs account for a small proportion in the image, which causes the model to lose the high-level semantic information of small objects during the continuous convolution.

Based on the above discussion, It is difficult to extract discriminative object features from the model trained by the existing methods. In addition, the existing small objects detection algorithms ignore the texture information and high-level semantic information of small objects, which limits the improvement of detection performance. So we proposed a knowledge-guided and information-rich model for small objects detection in foggy scenes. The model consists of three modules: knowledge guidance, texture information acquisition, and semantic information enrichment. Among them, the knowledge guidance module focuses on alleviating the adverse effects of foggy environment on detection performance; the texture information acquisition module is responsible for enriching the texture information of small

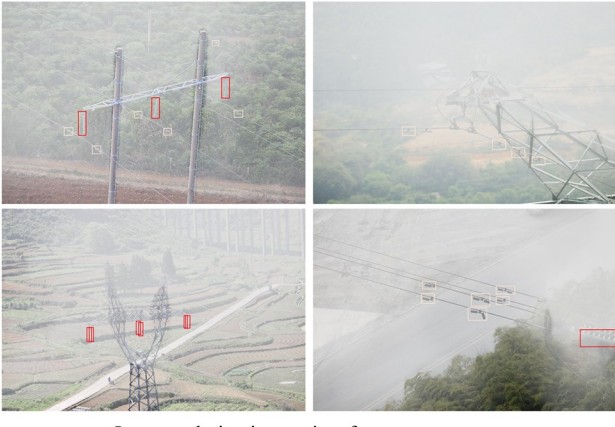 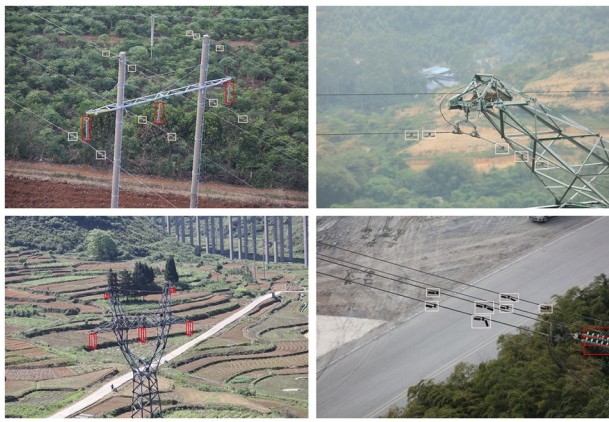

Low-resolution image  in a foggy scene | clear high-resolution image

**Fig 1. Low-resolution image (left) in a foggy scene and clear high-resolution image (right).**

objects in the feature map; the semantic information enrichment module obtains the high-level semantic information of small objects by constructing the attention mechanism between multi-scale feature maps.

To sum up, the contribution of this paper and the advantages of the proposed method are as follows:

1. In view of the difficulties caused by foggy scenes to the detection task, a knowledge guidance module is designed. It is used to extract the features of the clear image to assist the network to learn the foggy image, so that this model can better extract the discriminative object features.

2. Our designed a texture information acquisition module, which can enable the network to acquire the texture information of small objects from low-resolution images, and improve the model's representational ability to a certain extent.

3. In order to further complete the information of small objects, a semantic information enrichment module is proposed. In this module, a multi-scale feature attention mechanism is constructed to obtain the rich high-level semantic information of small objects. The validity and superiority of the proposed method are fully verified by experiments on two public data sets, "Cityscape to Foggy" and "CoCo".

## Related works

At present, CNN networks are generally used as feature extractors in object detection networks. In order to increase the receptive field, the CNN network will continuously shrink the feature map, which makes the feature information extracted from small objects originally containing very little information less or disappear. It is a huge challenge for detecting small objects in images. To address this problem, some researchers have proposed small objects detection based on multi-scale feature extraction, small objects detection assisted by high-resolution features, and small objects detection guided by content information.

### Small objects detection based on multi-scale feature extraction

Most of the existing object detection approaches are for general objects detection. Due to the loss of location information, the detection effect of this kind of method in small objects task has poor performance. In order to obtain the semantic information and position information of the object at the same time, small objects detection methods based on multi-scale feature extraction are proposed. This kind of methods extract features of different-scale images and fuses them to obtain rich object information. For example, Lin et al. [7] designed a top-down feature pyramid network, and merged the high-level semantic information and low-level location information obtained by the network, which significantly improves the performance of small objects detection. Given that deep-level features are difficult to retain the spatial and semantic information of small objects, Liu et al. [8] proposed an image pyramid guidance network and embedded it in each stage of the backbone network, greatly alleviating the information imbalance. In addition to the imbalance between semantic information and spatial information, the imbalance in training process is also a major contributing factor to the low performance of object detection. Thus, Pang et al. [9] proposed IoU sampling, balanced feature pyramid, and balanced L1 loss methods to respectively solve the sample imbalance, feature imbalance and objective function imbalance during the object detection and training.

Some researchers explored how to improve model efficiency while ensuring accuracy. For example, Tan et al. [10] proposed EfficientDet for object detection task. They designed a

weighted bi-directional feature pyramid network to integrate multi-scale features more efficiently and quickly, and proposed a hybrid scaling method to uniformly scale the width, depth and resolution of the network. Yet recently, it has been found that the feature pyramid networks can cause serious aliasing effects in the process of fusing feature maps. To solve this problem, Luo et al. [11] proposed an object detection method with enhanced channel information, which introduces a channel attention module to eliminate aliasing. In addition, they also proposed a subpixel jump fusion method, which effectively reduces the information loss in the process of channel reduction. This method only performs channel attention operations on each feature layer separately, and cannot address the problem of high-level semantic information loss of small objects. Considering that small objects have insufficient information in the original image, some researchers applied super-resolution to input image processing, and proposed a small objects detection method assisted by high-resolution features.

## Small objects detection assisted by high-resolution features

High-resolution features often contain rich texture information, which can play a positive role in improving the performance of small objects detection tasks. Therefore, some scholars put forward small objects detection methods that extract high-resolution features, and learn low-resolution images through its auxiliary model. Among them, the generative adversarial method can effectively align the features of different levels [12, 13]. Specifically, Li et al. [12] built a generator to map the feature of a small objects into a high-resolution feature. To ensure the accuracy of the mapping, a discriminator was designed to distinguish between high-resolution and low-resolution features. In this way, the network has the ability to extract high-resolution features from low-resolution images. Considering the category scores and position confidence involved in the object detection task, Bai et al. [13] introduced the idea of multi-task discrimination to design a multi-task discriminator. This discriminator can determine the categories and coordinates of the mapping features while distinguishing high and low resolutions. To further improve the super-resolution quality of small objects, Noh et al. [14] used high-resolution object features as a supervisory signal, and designed a generator to generate fine-grained high-resolution features.

For remote sensing images, Ji et al. [15] proposed an end-to-end method to integrate the generation and detection tasks to achieve vehicle detection. The generating networks are often prone to model collapse. To overcome this problem, Shermeyer et al. [16] proposed to use deep convolutional neural network (VDSR) or random forest super-resolution framework to perform super resolution processing on images of different levels and then train the object detection model. However, this approach is time-consuming as it needs to pre-train the super resolution model. The above problems can be addressed by an end-to-end training method. Such a method directly uses the high-resolution features as the supervision information and gives the model the ability to extract high-resolution image features from low-resolution ones by adversarial means.

## Small objects detection guided by content information

Content information guidance is a very popular research topic in object detection tasks in recent years. Excellent content information guided small objects detection algorithms have been proposed one after another. For example, Chen et al. [17] proposed two kinds of information at the image level and the relationship level between the objects, and designed a spatial memory network to store this information, and used this information to assist the model for detection in the iterative process. Early objects detection algorithms failed to consider the local and global information of the image at the same time. In response to this problem, Zhu et al.

[18] developed the global structure-local area network to extract the global features and local features of the image simultaneously for detection. In this process, the local features are obtained by the RoI pooling of position sensitivity. Once the model can learn the relationship between the objects, the detection effect can be enhanced to a certain extent. Therefore, Hu et al. [19] constructed an object relationship module which introduces the attention mechanism to describe the relationship between different objects.

Compared with general detection tasks, face detection is more challenging. This is mainly because of the small size, blurred quality and partial occlusion of the face in the image. To capture the contextual information in the image, Tang et al. [20] designed a new context anchor to extract high-level context features, and used a low-level feature pyramid network to integrate low-level information with them. To alleviate the mismatch between feature map resolution and receptive field size, Cao et al. [21] proposed a context information extraction module and an attention guidance module. The former is responsible for exploring and acquiring a large amount of contextual information, while the latter is to alleviate redundant context relationships and adaptively capture the dependencies between objects. Nevertheless, these methods are mostly designed for general object detection, and their detection performance on small objects is unsatisfactory. In addition, images acquired in real world are often impacted by weather conditions. For example, foggy images will result in poor image quality, thereby affecting the representational ability of the model.

In order to solve the above problems, this paper proposes a texture and semantic integrated small objects detection in foggy scenes. The model contains three modules: knowledge guidance, texture information acquisition, and semantic information enrichment. The three modules equip the model for small objects detection in foggy scenes.

## A texture and semantic integrated small objects detection model in foggy scenes

### Overview

The texture and semantic integrated small objects detection algorithm proposed in this paper consists of three modules. They are knowledge guidance, texture information acquisition, and semantic information enrichment, as shown in Fig 2. In the knowledge guidance module, the discriminative features extracted by the network from the clear image are used to guide the model to learn the foggy image. Different from low-resolution images, high-resolution ones often contain texture information of small objects. If the network can also extract features containing texture information from low-resolution images, it can improve the model's representational ability to a certain extent. To this end, a texture information acquisition module is designed. In this module, the features of high-resolution and low-resolution images are extracted separately. Next, adversarial learning is conducted between the two to align the features of low-resolution images with high-resolution features, so that the network can obtain texture information of small objects from low-resolution images.

In order to make the model trained on a clear image have a good effect in detecting foggy images, a knowledge guidance module is added to enable the feature extractor $E$ to extract discriminative object features. In order to enrich the small objects information on the feature map, the texture information acquisition module is added, and the training feature extractor $E$ can extract the feature map with rich texture information; for the three different scales feature map, because the small-scale and mesoscale features pass through more layers of convolution network, resulting in the loss of small objects semantic information. So the semantic information enrichment module is added to obtain the high-level semantic information of small objects. In this way, although the shallow features are convoluted continuously, the rich

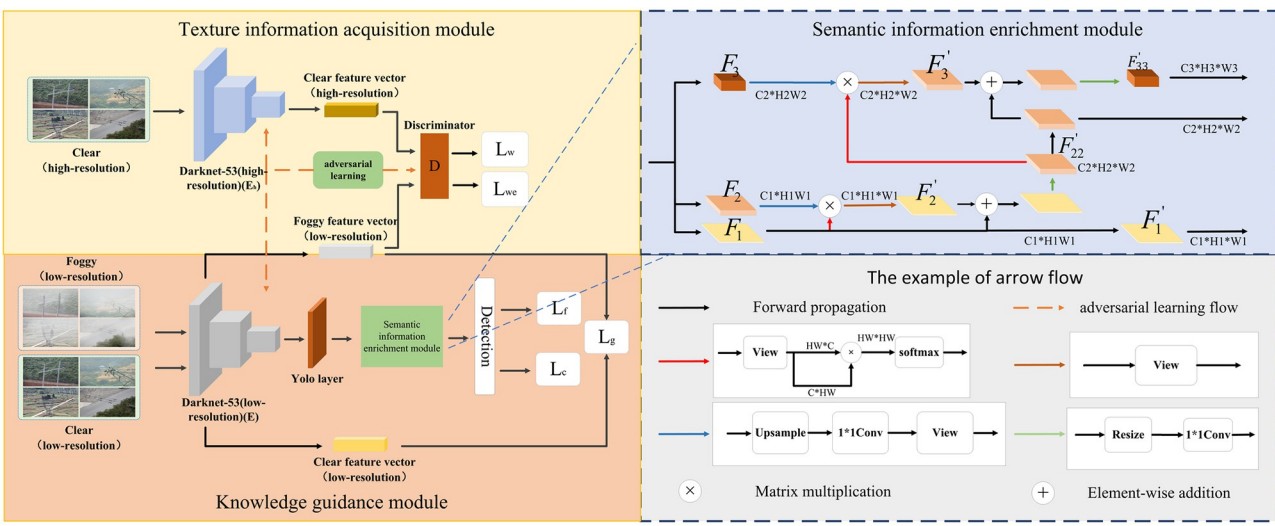

**Fig 2. The texture and semantic integrated small objects detection algorithm in foggy scenes.**

semantic information of small objects can still be retained. These three models will be introduced in detail below.

## Knowledge guidance module

In practice, many images are collected in foggy scenes. Such images have low contrast and sharpness, so it is difficult for the network to extract discriminative features of the object. To overcome this problem, a knowledge guidance module is proposed. In this module, the discriminative features extracted by the network from the clear image are used to guide the model to learn the foggy image. As shown in Fig 3, it is assumed that the low-resolution image in a foggy scene is $X = \{x_n\}_{n=1}^{N}$ and the clear low-resolution image is $X_c = \{x_n^c\}_{n=1}^{N}$, where $N$ is the number of images. First, the feature extractor $E$ is applied to extract the features $F$ and $F^c$ of foggy and clear low-resolution images. After that, $F$ and $F^c$ are fed into the Yolo layer to generate features of three scales, respectively denoted by $F_1, F_2, F_3$ and $F_1^c, F_2^c, F_3^c$. The dimensions of each scale feature map are respectively $C_1^*H_1^*W_1, C_2^*H_2^*W_2$ and $C_3^*H_3^*W_3$. It should be pointed out that $H_k = W_k(k = 1, 2, 3)$. Subsequently, $B$ candidate boxes are selected from each pixel of the feature map. For low-resolution images in a foggy scene, objects in the candidate frame are classified to ensure the discriminability of features. The category loss and category confidence loss are as follows:

$$L_{cls}(E) = -\sum_{i=0}^{S^2}\sum_{j=0}^{B}I_{i,j}^{obj}\sum_{c\in class}p_i(c)log(\hat{p}_i(c)) \tag{1}$$

$$L_{conf}(E) = \sum_{i=0}^{S^2}\sum_{j=0}^{B}I_{i,j}^{obj}(c_i - \hat{c}_i)^2 + \sum_{i=0}^{S^2}\sum_{j=0}^{B}I_{i,j}^{noobj}(c_i - \hat{c}_i)^2 \tag{2}$$

Here, $S$ is the dimension of the feature map, i.e., $H_1, H_2$ and $H_3$. $I_{i,j}^{obj}$ is the indicator function, which means that the object is in the candidate box $j$ at the position $i$ on the feature map. $p_i(c)$ is the real category label, and $\hat{p}_i(c)$ is the predicted result. $c_i$ and $\hat{c}_i$ are the confidence of the real and predicted categories, respectively.

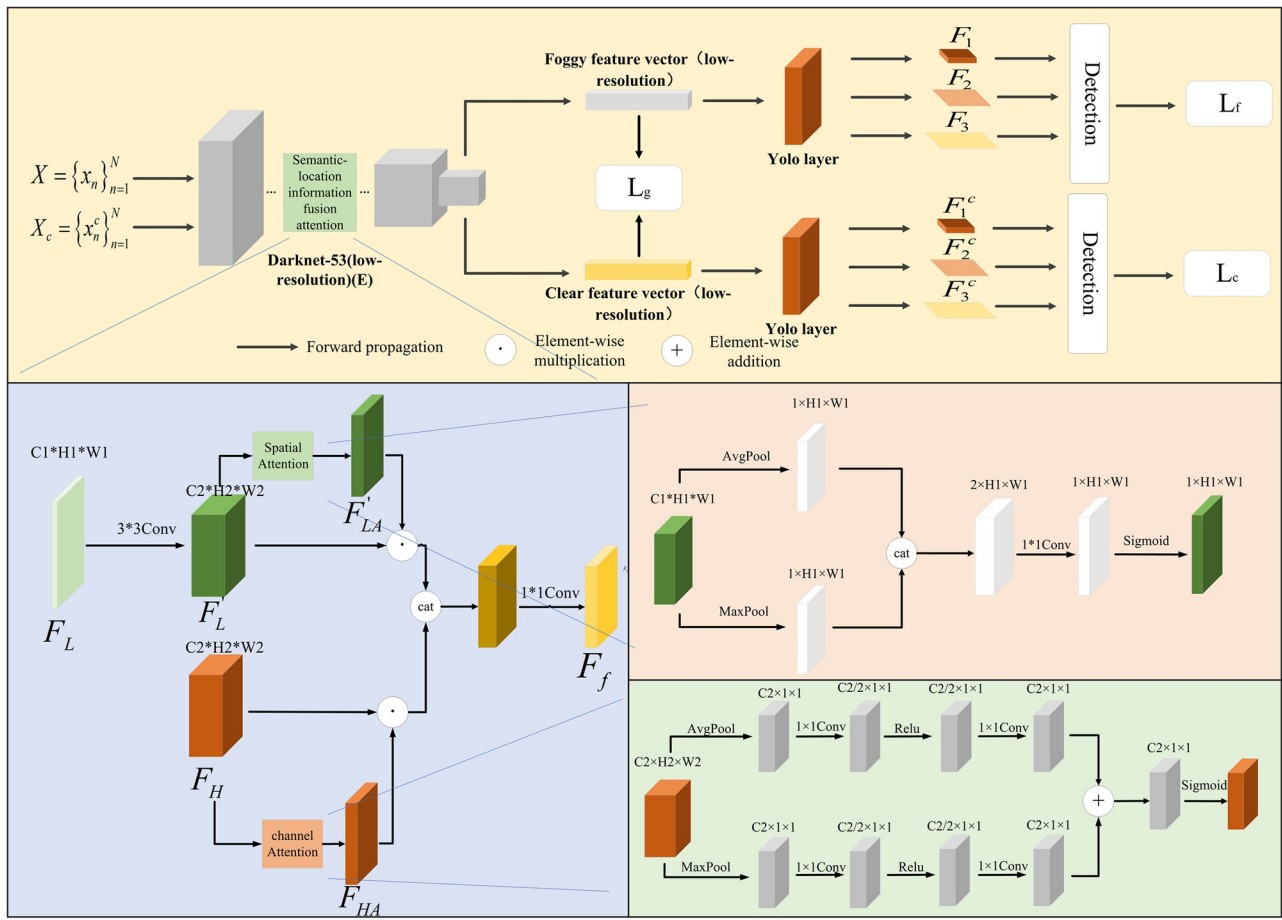

**Fig 3. Knowledge guidance module.**

In addition to the above two losses, the detection task also needs to determine the coordinates of the object in the picture. The corresponding loss is:

$$L_{coord}(E) = \sum_{i=0}^{S^2}\sum_{j=0}^{B} I_{i,j}^{obj}(2 - w_i \times h_i)[(x_i - \hat{x}_i)^2 + (y_i - \hat{y}_i)^2 + (h_i - \hat{h}_i)^2 + (w_i - \hat{w}_i)^2] \quad (3)$$

Where $x_i, y_i, w_i$ and $h_i$ respectively represent the center point position and scale in the real box, $\hat{x}_i, \hat{y}_i, \hat{w}_i$ and $\hat{h}_i$ are the the center point position and scale in the prediction box.

Therefore, for foggy low-resolution images, the detection loss is:

$$L_f(E) = \sum_{F_1,F_2,F_3} L_{cls} + L_{conf} + L_{coord} \quad (4)$$

Similarly, to ensure the discriminability of clear image features, loss constraint is performed on $F_1^c$, $F_2^c$ and $F_3^c$. There is:

$$L_c(E) = \sum_{F_1^c,F_2^c,F_3^c} L_{cls} + L_{conf} + L_{coord} \quad (5)$$

The features extracted by the network from clear images tend to be more discriminative. In order to give the model the ability to extract discriminative features from foggy images, a knowledge guidance mechanism is designed. In this mechanism, the strategy of knowledge distillation is adopted to promote the self-learning of the feature encoder $E$. There is:

$$L_g(E) = KL(F, F^c) \qquad (6)$$

Where KL is the KL divergence, which can be expressed as:

$$KL(F, F^c) = \sum_{r=1}^{R}[p(F_r)\log p(F_r) - p(F_r)\log p(F_r^c)] \qquad (7)$$

$p(\bullet)$ means to convert the feature map to a vector and use softmax function to process it, and $R$ is the number of pixels in the feature map.

In order to make feature extractor $E$ extracts deep features have stronger position representation; we designed a semantic position information fusion attention module, as shown in Fig 3. In this module, the third layer feature $F_L$ extracted by the feature extractor is convoluted by $3 * 3$ to generate the feature $F_L'$, which is consistent with the size of the ninth layer feature. Because the third layer features pass through few convolution layers and have rich location information, the spatial attention operation is carried out on $F_L'$ to generate image feature $F_{LA}'$, so that the network can pay attention to the interest region in the third layer image feature. Because the ninth layer features undergo multi-layer convolution, the semantic information of the feature map is very rich. Channel attention operation is performed on $F_H$ to generate the feature map $F_{HA}$, so that the network can pay attention to the interest semantic information. In order to enrich the position representation of $F_{HA}$, $F_{LA}'$ and $F_{HA}$ are spliced and fused, and then through $1 * 1$ convolution, $F_{HA}$ obtains the position information in $F_{LA}'$. The Eq (8) is as follows:

$$F_f = CONV(Cat(F_L' \odot F_{LA}', F_H \odot F_{HA})) \qquad (8)$$

Where, $\odot$ represents element level multiplication, $Cat$ represents splicing in channel dimension, $CONV$ represents $1 * 1$ convolution. Through this model processing, the feature extractor $E$ can extract deep features with rich location information from the foggy image.

## Texture information acquisition module

As discussed earlier, high-resolution images often contain the texture information of the object. If the network can also obtain such information in low-resolution images, the model's characterization ability can be further improved. To achieve this goal, a texture information acquisition module is designed. This module contains two feature extractors $E$ and $E_h$, and one discriminator $D$, as shown in Fig 4. Among them, the function of $E$ is consistent with that in the knowledge guidance module, and it is used to extract the features of low-resolution foggy images; $E_h$ is used to extract the features of high-resolution images in foggy scenes; the discriminator judges whether the extracted features are from high-resolution images or low-resolution images. For example, given a high-resolution image $X_h = \{x_i^h\}_{i=1}^N$, the feature extractor $E_h$ is applied to extract the feature map $F^h$, and then adversarial learning is adopted to

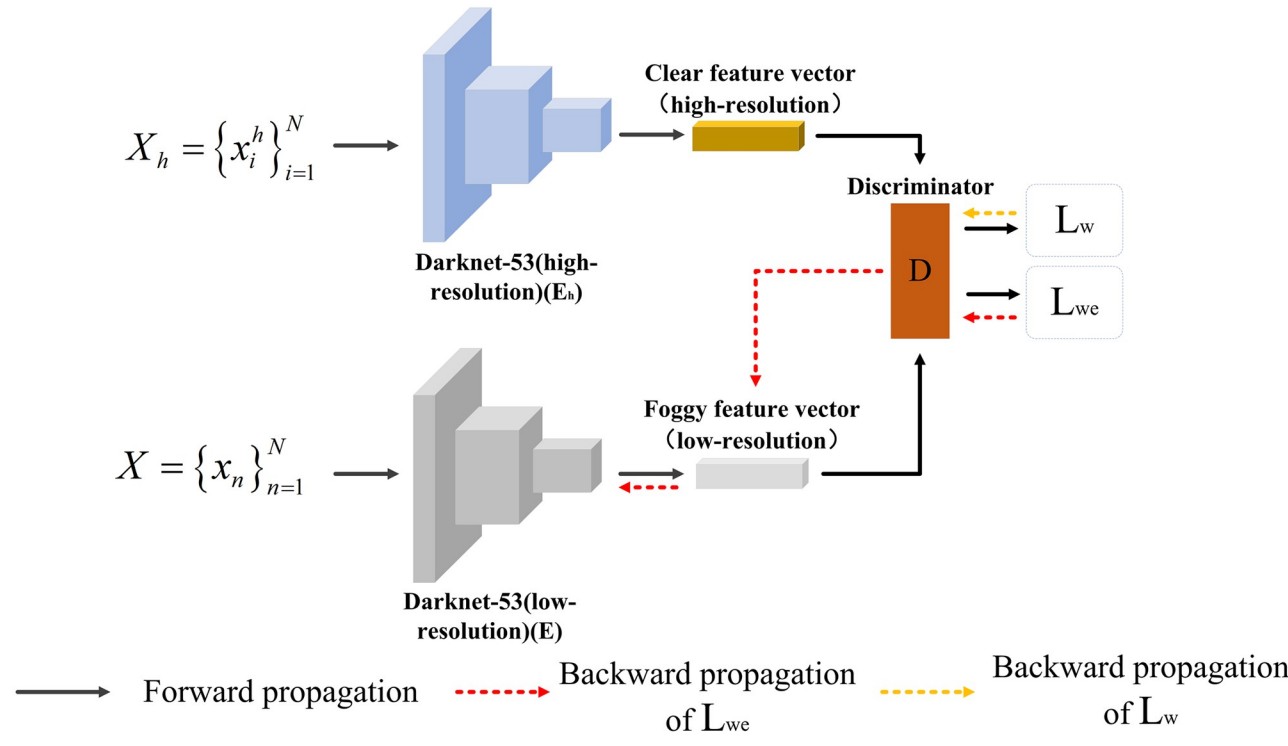

**Fig 4. Texture information acquisition module.**

constrain the feature extractor $E$ and the discriminator $D$.

$$L_w(E, D) = \mathrm{E}_{x_h \sim p_{\mathrm{data}}(x_h)}[\log D(F^h)] + \mathrm{E}_{x \sim p_{\mathrm{data}}(x)}[\log(1 - D(E(x)))] \tag{9}$$

$$L_{we}(E, D) = E_{x_h \sim p_{\mathrm{data}}(X_h)}[\log(1 - D(F^h))] + E_{x \sim p_{\mathrm{data}}(x)}[\log(D(E(x)))] \tag{10}$$

Here, $E(x)$ is the feature extracted by feature extractor $E$ from image $x$.

After the above operations, the feature extractor now has the ability to extract high-resolution features, enriching the texture information of the small objects.

## Semantic information enrichment module

Feature maps of different scales contain different types of object information, and continuous convolution operations will cause the loss of high-level semantic information of small objects. To overcome this problem, this paper designs a semantic information enrichment module to obtain the high-level semantic information of small objects. As shown in Fig 5, the feature extractor $E$ extracts the features $F_1$, $F_2$ and $F_3$ of low-resolution foggy images. In order to retain the key information of the object in the feature map, a self-attention mechanism is constructed on the feature map of each scale. Specifically, the feature maps of each scale are processed by resizing to change their dimensions to $C_1^*(H_1 \ W_1)$, $C_2^*(H_2 \ W_2)$ and $C_3^*(H_3 \ W_3)$ Subsequently, the feature maps are processed by transposition, multiplication, and normalization to get the

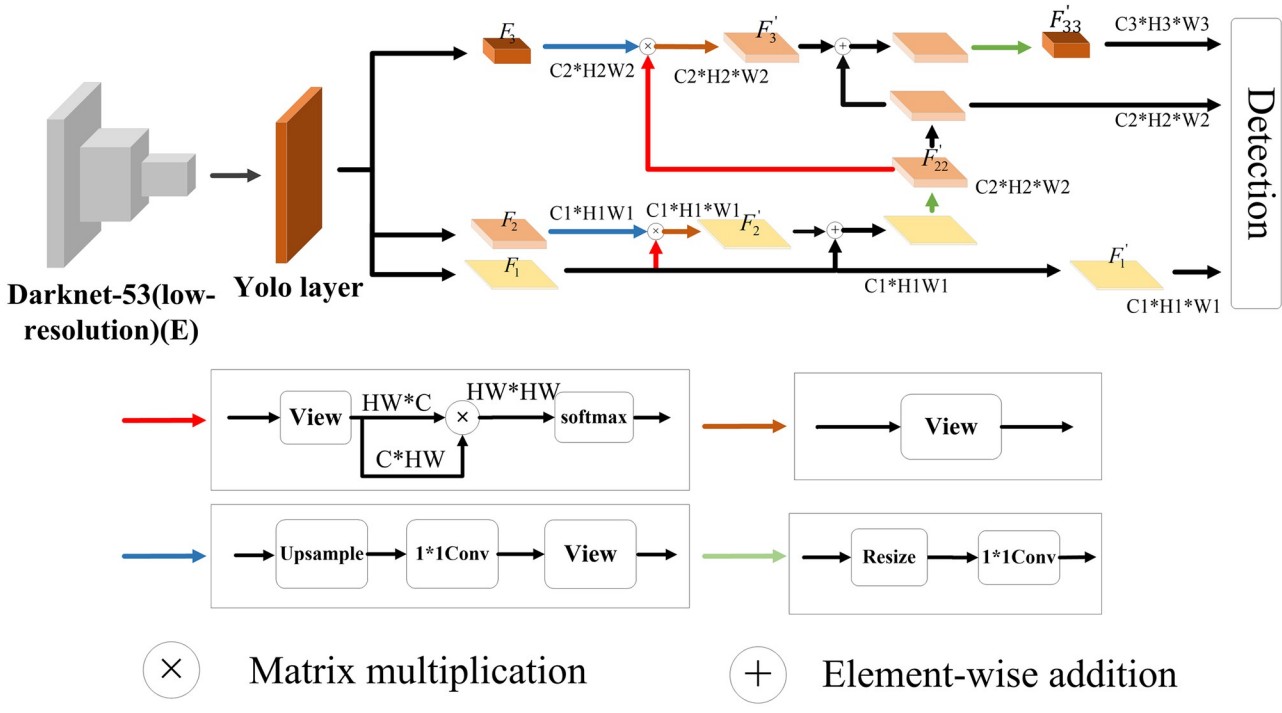

**Fig 5. Semantic information rich module.**

attention matrices:

$$A_1 = \text{softmax}(F_1^T F_1) \tag{11}$$

$$A_2 = \text{softmax}(F_{22}'^T F_{22}') \tag{12}$$

Next, by transposing and multiplying the feature map of each scale with the corresponding attention matrix, we get:

$$F_2' = F_2 A_1^T \tag{13}$$

$$F_3' = F_3 A_2^T \tag{14}$$

$F_1'$, $F_2'$ and $F_3'$ are the feature maps after self-attention operation, and such feature maps contain the key information of the object on feature maps of each scale.

Unlike $F_1'$, the scales of $F_2'$ and $F_3'$ are small owing to continuous convolution operations, so the high-level semantic information of small objects is often lost. To retain this information, a feature fusion mechanism is constructed on feature maps of different scales. To ensure that $F_2'$ contains the information of small objects in $F_1'$, the features of the two scales are fused to get:

$$A_{12} = F_1 \oplus F_2' \tag{15}$$

Where $A_{12}$ is the feature after fusion of $F_1$ and $F_2'$, and its dimension is $C_1 {}^* H_1 {}^* W_1$. This operation introduces the key object information in $F_1'$ to $F_2'$. On this basis, the fused features

are subjected to up-sampling and convolution to get:

$$F'_{22} = \text{Conv}(\text{Upsample}(A_{12}))\tag{16}$$

Where the up-sampling multiple is 2, and conv is $1^*1$ convolution. By this way, the small objects information of feature map $F'_1$ is introduced into $F'_{22}$. It should be pointed out that $F'_{22}$ and $F_2$ have the same dimension of $C_2^*H_2^*W_2$. Similarly, to make $F'_3$ contain the information in $F'_{22}$, the above operations are performed on $F'_{22}$. The first is to calculate the self-attention map of $F'_{22}$ and then to obtain the channel attention matrix $A_{23} = F'_{22} \oplus F'_3$. Subsequently, the fused feature $A_{23}$ are subjected to up-sampling and convolution to get $F'_{33} = \text{Conv}(\text{Upsample}(A_{23}))$. This feature map has the same dimension of $F'_3$.

After the above operations, the deep feature maps $F'_{22}$ and $F'_{33}$ can contain the small objects information in the feature map $F'_1$. Then category loss, confidence loss and coordinate position loss are performed on the obtained feature maps $F'_1$, $F'_{22}$ and $F'_{33}$ to get:

$$L_s(E) = \sum_{F'_1, F_{22}, F'_{33}} L_{cls} + L_{conf} + L_{coord}\tag{17}$$

## Total loss

In summary, the total loss of the small objects detection algorithm proposed in this paper is:

$$L_{total}\ (E, E_c, E_h, D) = L_f(E) + L_c(E_c) + \lambda_1 L_g(E, E_c) + \lambda_2 L_w(E, D) + L_{we}(E, D)\tag{18}$$

Where $\lambda_1$ and $\lambda_2$ are hyper-parameters, which are used to weigh the proportions of the loss items $L_g(E, E_c)$ and $L_w(E, D)$ in the total loss.

**Algorithm 1** the proposed algorithm.

```
Definition: two feature encoders E_h and E; one discriminator D
Input:
Labeled high-resolution clear image X_h = {x_i^h}_{i=1}^N, labeled low-resolution
clear image X_c = {x_n^c}_{n=1}^N, label P = {p_n^c}_{n=1}^{N_c}, and unlabeled low-resolution
foggy image X = {x_n}_{n=1}^N
Output:
optimal model E*
Image preprocessing:
Scale the input image of any size to 608*608*3
Optimize:
for each k ∈ [1, n] do:
 (a) Input image X_h = {x_i^h}_{i=1}^N into E_h feature F_i^h and input images X = {x_n}_{n=1}^N
 and X_c = {x_n^c}_{n=1}^N into E to get features F_i and F_i^c;
 (b) Input features F_i^h and F_i^c into discriminator D and then train E_h and
 D with formulas (9) and (10);
 (c) Input features F_i^c and F_i into KL loss and train E with formula (7);
 (d) Input feature F_i into Yolo layer and train E with formula (4);
end for
```

## Experiments

### Data set and evaluation indicators

In order to verify the effectiveness and superiority of the proposed algorithm, a large number of experiments were carried out on two available large-scale data sets, i.e. "Cityscape to Foggy" [22] and "CoCo" [23], and they were compared with some of the most advanced methods. "Cityscape to Foggy" is mainly used for semantic segmentation tasks. Cityscape dataset had 3457 images, including 2965 images in the training set and 492 in the test set; FoggyCityscape

dataset had 3457 images, of which 2965 were in the training set and 492 in the test set. Therefore, eight categories of data-"Bus", "Bicycle", "Car", "Motorcycle", "Person", "Rider", "Train" and "Truck" were singled out from the "Cityscape to Foggy" dataset, and their labels were converted into formats suitable for object detection tasks. By contrast, the "CoCo" data set had 123287 fog-free images, of which 118287 in the training set and 5000 were in the test set. To verify that the algorithm proposed in this paper can effectively detect small objects in foggy scenes, the foggy algorithm [24] was applied to fog the COCO data set, and 123287 foggy images (118287 in the training set and 5000 in the test set) were obtained. All experiments in this paper used average prediction accuracy ($mAP$) as the evaluation index, and the formula is shown in Eq (19).

$$mAP = \frac{1}{C}\sum_{c}\sum_{k=1}^{N}\max_{\tilde{k}\geq k}P(\tilde{k})\Delta r(k) \tag{19}$$

Where $C$ is the total number of categories, $N$ the sample size in one category, $\max_{\tilde{k}\geq k}P(\tilde{k})$ is the accuracy of the maximum confidence greater than the confidence threshold, $\Delta r(k)$ is the change in the recall rate when the sample size changes from $k-1$ to $k$. On the coco data set, the following evaluation indicators are derived based on mAP: $mAP_{50}$ represents the average prediction accuracy of 0.5 for the intersection ratio of the target's prediction frame and its real frame; $mAP_{75}$ represents the average prediction accuracy of 0.75 for the intersection ratio of the target's prediction frame and its real frame; $mAP_{S}$ means the average prediction accuracy of the target area less than $32^*32$ pixels; $mAP_{M}$ means the average prediction accuracy of the target area greater than $32^*32$ but less than $96^*96$ pixels; $mAP_{L}$ means the average prediction accuracy of the target area larger than $96^*96$ pixels.

## Experimental details

The experiment was performed on a GTX 3090Ti graphics card. The feature encoders in this paper all used the Darknet-53 network as the feature extractors. $E, E_c$ and $E_h$ did not share parameters with each other. The discriminator $D$ consists of three convolution-pooling-normalization blocks and one fully connected layer. During the experiment, the images were pre-processed by random flip, random fill, and random crop, and the size of the image was uniformly scaled to 608608. The model was trained for 100 generations, the batch size was 2, and the initial learning rate was 0.007. The learning rate adjustment algorithm adopted the cosine annealing algorithm.

In the test, only the feature encoder $E$ was used to extract the features of low-resolution images in foggy scenes for prediction.

## Method comparison

In this section, the proposed algorithm is compared with some of the most advanced methods [25–32] on "Cityscape to Foggy" and "CoCo" data sets to show its advantages. On "Cityscape to Foggy" the detection performance of eight categories of "Bus", "Bicycle", "Car", "Motorcycle", "Person", "Rider", "Train" and "Truck" are compared. The results are listed in Table 1.

It can be seen from Table 1 that compared with other object detection algorithms, the algorithm proposed in this paper performs the best under foggy conditions. The proposed algorithm is also compared with the state-of-the-art methods on the foggy "CoCo" data set, as shown in Table 2. It can be seen that the algorithm proposed has the best performance.

**Table 1. Experimental results of different methods on "Cityscape to Foggy" data set.**

| Methods | mAP |
|---|---|
| CPM-R-CNN [25] | 25.8 |
| Libra R-CNN [26] | 31.9 |
| PPA [27] | 34.0 |
| Vfnet [28] | 31.2 |
| Paa [29] | 26.5 |
| Gfl [30] | 30.7 |
| Sabl [31] | 23.8 |
| CST-DA [32] | 32.1 |
| Ours | **46.2** |

**Table 2. Experimental results of different methods on the foggy "CoCo" data set.**

| Methods | mAP | $mAP_{50}$ | $mAP_{75}$ | $mAP_S$ | $mAP_M$ | $mAP_L$ |
|---|---|---|---|---|---|---|
| Libra R-CNN [26] | 30.5 | 38.6 | 32.0 | 14.0 | 35.7 | 42.1 |
| PPA [27] | 32.7 | 41.9 | 36.2 | 14.0 | 35.8 | 48.2 |
| Vfnet [28] | 32.9 | 48.3 | 35.2 | 14.3 | 36.0 | 48.0 |
| Paa [29] | 32.6 | 48.1 | 34.7 | 11.9 | 35.4 | 51.1 |
| Gfl [30] | 28.3 | 43.3 | 29.9 | 10.6 | 30.4 | 44.1 |
| Sabl [31] | 29.8 | 45.7 | 31.4 | 10.5 | 32.6 | 46.3 |
| Ours | **33.3** | **49.9** | 35.9 | **14.4** | **38.2** | 47.7 |

## Ablation experiment

In order to prove the effectiveness of each module in the proposed algorithm, a series of ablation experiments were carried out. In this process, the model with all three modules removed was used as the benchmark, that is, only the detection loss constraint model for low-resolution images in foggy scenes was used. As shown in Table 3, the model using only benchmark training achieved an average prediction accuracy of 39.4% on "Cityscape to Foggy".

Knowledge guidance module: It can be seen from Table 3 that when the knowledge guidance module is added, the performance is greatly improved compared with the benchmark, by a mean predication accuracy of 4.7% on "Cityscape to Foggy". It indicates that the proposed

**Table 3. Ablation experimental results of the model on the "Cityscape to Foggy" dataset.**

| - | Cityscape to Foggy |
|---|---|
| Methods | mAP |
| Benchmark | 39.4 |
| Benchmark + knowledge guidance | 44.1 |
| Benchmark + semantic information enrichment | 39.6 |
| Benchmark + texture information acquisition | 40.9 |
| Benchmark + semantic information enrichment + texture information acquisition | 42.8 |
| Benchmark + knowledge guidance + texture information acquisition | 44.7 |
| Benchmark + knowledge guidance + semantic information enrichment | 44.3 |
| Benchmark + knowledge guidance + texture information acquisition + semantic information enrichment(ours) | 46.2 |

knowledge guidance module can effectively alleviate the adverse effects of foggy environment on recognition performance.

Texture information acquisition module: When the texture information acquisition module is added to the model, the detection performance is further improved. The mean prediction accuracy on the "Cityscape to Foggy" dataset increases from 39.4% to 40.9%. This is mainly because the proposed texture information acquisition module enriches the texture information in the features and improves the representation ability of the model.

Semantic information enrichment module: The loss of high-level semantic information is partly to blame for the limited detection performance. Thus, when the semantic information enrichment module is added to the model, the detection performance is improved. The mean prediction accuracy on "Cityscape to Foggy" dataset increases from 39.4% to 39.6%. It is therefore can be concluded that the proposed semantic information enrichment module plays a positive role in enhancing detection performance.

It can be seen from Table 3 that after considering the three modules Knowledge guidance module, Texture information acquisition module and Semantic information enrichment module at the same time, the model achieves a better performance. The mean prediction accuracy on the "Cityscape to Foggy" dataset reached 46.2%.

## Parameter analysis

In the algorithm proposed in this paper, $\lambda_1$ and $\lambda_2$ are hyper-parameters, which are used to weigh the proportions of loss terms $L_g(E, E_c)$ and $L_w(E, D)$ in the total loss respectively. To demonstrate that the parameter value is the optimal, the parameter analysis experiment was carried out, and the experimental results are shown in Fig 6 below. It should be noted that when one parameter is analyzed, the other parameter is fixed.

The impact of $\lambda_1$ on model performance: Parameter $\lambda_1$ is used to control the relative weight of the loss term $L_g(E, E_c)$ in the total loss. As shown in Fig 6(a), when $\lambda_1$ ranges within [0.6,1.0], mAP is on the rise; it reaches the maximum when $\lambda_1 = 1.0$ and starts to decline thereafter. This indicates that the optimal value of $\lambda_1$ is 1.0.

The impact of $\lambda_2$ on model performance: Parameter $\lambda_2$ is used to control the relative weight of $L_w(E, D)$. It can be observed from Fig 6(b) that as $\lambda_2$ rises from 0.6 to 0.8, the detection

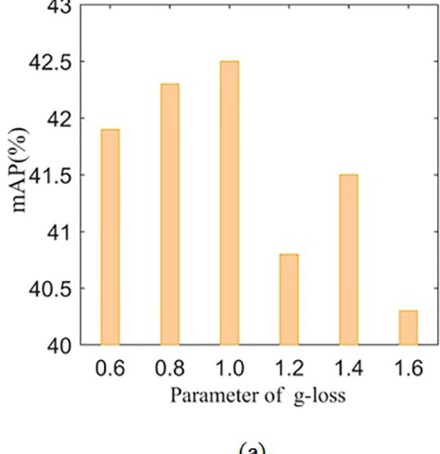

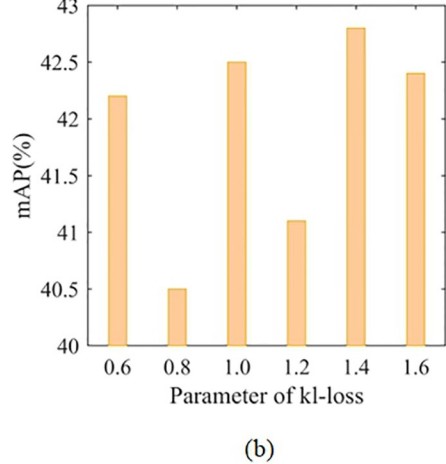

(a)                                                       (b)

**Fig 6.** Parameter analysis: (a) shows $L_g$ in the total loss, the influence of different weights on the total loss, and the weight is recorded as $\lambda_1$; (b) shows $L_w$ in the total loss, the influence of different weights on the total loss, and the weight is recorded as $\lambda_2$.

performance shows a negative growth, and varies greatly when $\lambda_2$ ranges from 0.6 to 1.6. At $\lambda_2 = 1.4$, mAP reaches its maximum. So the optimal value of $\lambda_2$ is 1.4.

## Visual analysis

In order to better prove the effectiveness of this method, this paper uses yolov5 method and this method to carry out visual display on the cityscape to foggy dataset, as shown in Fig 7.

As can be seen from Fig 7, this method can not only well detect targets in foggy scenes, but also detect small targets in foggy scenes. It can be proved that this method not only solves the

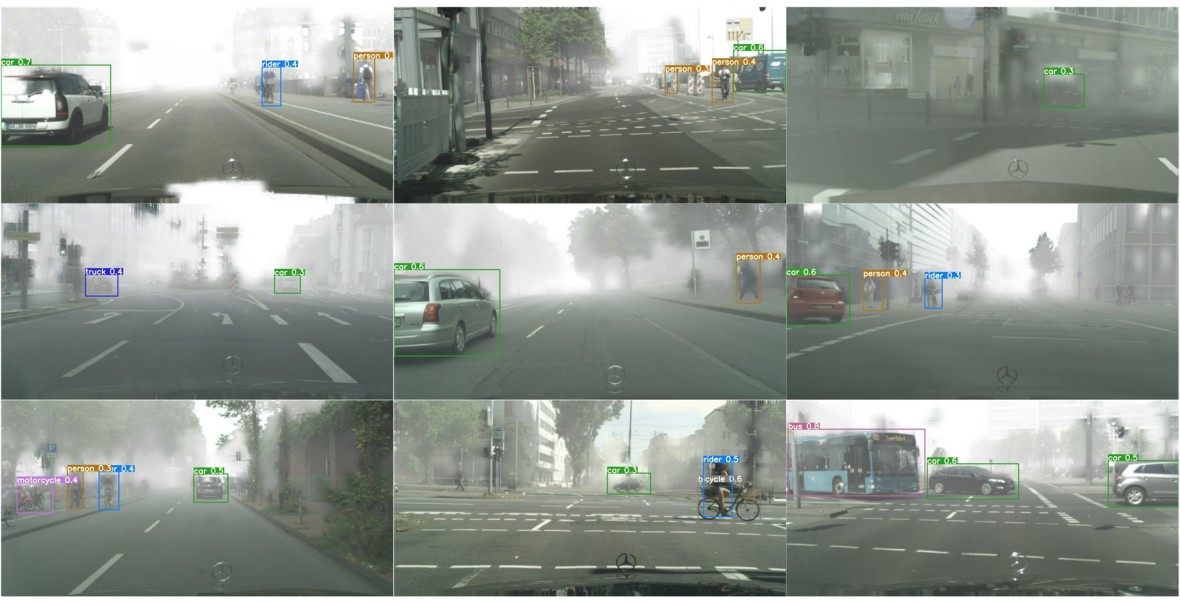

(a) Yolov5 method

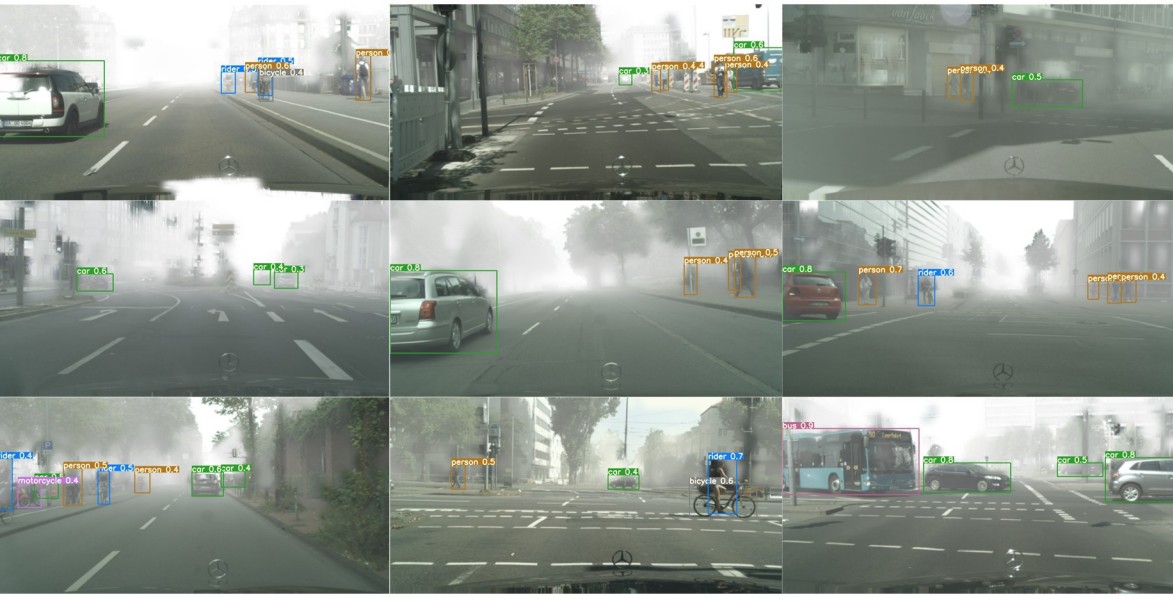

(b) Our method

**Fig 7. Visualization results on the Cityscape to Foggy dataset.**

problem of domain migration between different data sets, but also can effectively extract the information of small targets in shallow features and improve the ability of model to detect small targets.

## Conclusions

In this paper, we study the small objects detection task in foggy scene. It is difficult to extract discriminative object features from the model trained by the existing methods. In addition, the existing small objects detection algorithms ignore the texture information and high-level semantic information of small objects, which limits the improvement of detection performance. Aiming at the above problems, this paper proposes a Texture and semantic integrated small objects detection algorithm in foggy scenes. Specifically, the discriminant features extracted by the model from clear images are used to guide the model to learn, which alleviates the negative impact of foggy images on the detection performance. Given that the existing algorithms are difficult to obtain the texture information and high-level semantic information of small objects, the adversarial learning strategy is adopted to give the network the ability to obtain the texture information of small objects from low-resolution images. At the same time, a multi-scale feature map attention mechanism is constructed to further enrich the high-level semantic information of small objects. The effectiveness and superiority of the method proposed in this paper have been fully verified by a series of experiments.

## Supporting information

**S1 Dataset.**
(DOCX)

## Author Contributions

**Methodology:** Zhengyun Fang, Hongbin Wang.

**Project administration:** Shilin Li.

**Software:** Yi Hu, Xingbo Han.

**Writing – original draft:** Hongbin Wang, Yi Hu.

**Writing – review & editing:** Hongbin Wang.

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
