## [Decision Letter · Decision Letter 0]

3 Nov 2021

PONE-D-21-32064Knowledge-guided and information-rich small target detection in foggy scenesPLOS ONE

Dear Dr. Wang,

Thank you for submitting your manuscript to PLOS ONE. After careful consideration, we feel that it has merit but does not fully meet PLOS ONE’s publication criteria as it currently stands. Therefore, we invite you to submit a revised version of the manuscript that addresses the points raised during the review process. As you will infer from below that there was a disagreement between the reviewers regarding enthusiasm for this work. Reviewer 3 was of the view that manuscript partly describes a technically sound piece of work and recommended reject. On the other hand, reviewer 1 and 2  had made certain observations to improve your work and recommended minor revisions. After considering comments of reviewers, the editors decision is "major revision". Please incorporate all comments raised by reviewers.

We look forward to receiving your revised manuscript.

Kind regards,

Gulistan Raja

Academic Editor

PLOS ONE

Journal Requirements:

2.  Please ensure that you refer to Figure 6 in your text as, if accepted, production will need this reference to link the reader to the figure.

3. We note that Figures 1 and 2 in your submission contain [map/satellite] images which may be copyrighted. All PLOS content is published under the Creative Commons Attribution License (CC BY 4.0), which means that the manuscript, images, and Supporting Information files will be freely available online, and any third party is permitted to access, download, copy, distribute, and use these materials in any way, even commercially, with proper attribution. For these reasons, we cannot publish previously copyrighted maps or satellite images created using proprietary data, such as Google software (Google Maps, Street View, and Earth). For more information, see our copyright guidelines: http://journals.plos.org/plosone/s/licenses-and-copyright.

a. You may seek permission from the original copyright holder of Figures 1  and 2 to publish the content specifically under the CC BY 4.0 license.  

Reviewers' comments:

Reviewer's Responses to Questions

**Comments to the Author**

1. Is the manuscript technically sound, and do the data support the conclusions?

Reviewer #1: Yes

Reviewer #2: Yes

Reviewer #3: Partly

2. Has the statistical analysis been performed appropriately and rigorously? 

Reviewer #1: Yes

Reviewer #2: Yes

Reviewer #3: N/A

3. Have the authors made all data underlying the findings in their manuscript fully available?

Reviewer #1: Yes

Reviewer #2: Yes

Reviewer #3: Yes

4. Is the manuscript presented in an intelligible fashion and written in standard English?

Reviewer #1: Yes

Reviewer #2: Yes

Reviewer #3: No

5. Review Comments to the Author

Reviewer #1: The manuscript titled “Knowledge-guided and information-rich small target detection in foggy scenes” combines texture information and semantic information to detect small target in the foggy images. The work is interesting and the result support the conclusion well. However there are some flaws necessary to be polished to improve the quality of this manuscript. The novelty of the research is not outstandingly presented; the English language should be improved either.

Reviewer #2: In general, this paper is well written, but I have to point out that there are still some grammar and word mistake.

1.The author should polish the language of the paper.

2.It is suggested that the paper should be revised in strict accordance with the required format. For example: the position marked above the author's name and affiliations; each paragraph is indented and so on.

3.The recommended font must meet the requirements for the template.

4.The main motivation and contribution should be clearly highlighted in the introduction section.

5.The conclusion of the article is too simple, it should be described more fully to illustrate the feasibility and advantages of the method in this article.

6.It is recommended that the requirements of references and templates be strictly unified.

7.The naming format of the diagram is checked against the template.

8.As shown in Figure 2, the description of the algorithm should not be in the form of screenshots. It is recommended to replace it with Table.

9.The arrow flow of each diagram is recommended to be described in colloquial language for easy reading.

10.It is recommended to specify the color of the arrow and the meaning of the dotted and solid lines in each diagram.

11.The content and conclusion of the study are too simple, so I suggest to enrich the content of you work.

In order to make this article more fluent and easier to understand, I suggest the authors consider about those above eleven problems.

Reviewer #3: (1) Consider improving and adding a more suitable knowledge distillation model structure for application. The part of the ablation experiment is just the split and combination of each module, and more adequate experiments can be considered.

(2) Two feature extractors E are used to obtain Eh. Is it redundant? Can it be replaced by a general feature extractor, with two different resolutions as input, so that E and D confront each other;

(3) The algorithm flow is not clear enough, the algorithm format is wrong, it should not be a screenshot;

(4) English translation has grammatical errors or has not been investigated, such as target detection .

(5) There is a lack of detailed process description below the paper frame diagram.

6. PLOS authors have the option to publish the peer review history of their article (what does this mean?). If published, this will include your full peer review and any attached files.

Reviewer #1: No

Reviewer #2: No

Reviewer #3: No

---

## [Author Response · Author response to Decision Letter 0]

23 Mar 2022

Thank you for allowing a resubmission of our manuscript, with an opportunity to address the reviewers’ comments.

Journal Requirements:

Case1: Please ensure that your manuscript meets PLOS ONE's style requirements, including those for file naming. 

Response: We revised this manuscript accordance with the PLOS ONE's style requirements.

Case2: Please ensure that you refer to Figure 6 in your text as, if accepted, production will need this reference to link the reader to the figure.

Response: We revised Figure 6 according to the algorithm’ style requirements, such as algorithm 1.

Case3: We note that Figures 1 and 2 in your submission contain [map/satellite] images which may be copyrighted. 

Response: We revised them. Such as Figures 1 and 2. The pictures in Figure 1 and Figure 2 are taken by us, and there is no copyright problem.

Reviewer #1:

Case1: The manuscript titled “Knowledge-guided and information-rich small target detection in foggy scenes” combines texture information and semantic information to detect small target in the foggy images. The work is interesting and the result support the conclusion well. However there are some flaws necessary to be polished to improve the quality of this manuscript. The novelty of the research is not outstandingly presented; the English language should be improved either.

Response: The quality of this manuscript has been improved, the existing English language problems have been improved, and the novelty of the research is also improved.

Reviewer #2:

Case1: The author should polish the language of the paper.

Response: The existing English language problems have been improved.

Case2: It is suggested that the paper should be revised in strict accordance with the required format. For example: the position marked above the author's name and affiliations; each paragraph is indented and so on.

Response: We revised this manuscript accordance with the required format.

Case3: The recommended font must meet the requirements for the template.

Response: I rearranged the typesetting according to the PLOS ONE's Latex template.

Case4: The main motivation and contribution should be clearly highlighted in the introduction section.

Response: We revised this manuscript in the introduction section.

Case5: The conclusion of the article is too simple, it should be described more fully to illustrate the feasibility and advantages of the method in this article.

Response: We supplemented the conclusion of the article.

Case6: It is recommended that the requirements of references and templates be strictly unified.

Response: We revised it according to the requirements of references and templates.

Case7: The naming format of the diagram is checked against the template.

Response: We revised it.

Case8: As shown in Figure 2, the description of the algorithm should not be in the form of screenshots. It is recommended to replace it with Table.

Response: We revised it according to the algorithm’ style requirements, such as algorithm 1.

Case9: The arrow flow of each diagram is recommended to be described in colloquial language for easy reading.

Response: Thank you for your suggestions. In the revision manuscript, we have added arrow examples to facilitate readers' reading.

Case10: It is recommended to specify the color of the arrow and the meaning of the dotted and solid lines in each diagram.

Response: Thank you for your suggestions. In the revision manuscript, we have added arrows of different colors and the meaning of points.

Case11: The content and conclusion of the study are too simple, so I suggest to enrich the content of you work.

Response: Thank you very much for your suggestions. We added the semantic location information fusion attention module to make the deep features extracted by the feature extractor have stronger location representation, so as to assist the semantic information enhancement module and enrich the location information of large-scale features.

Reviewer #3: 

Case1: Consider improving and adding a more suitable knowledge distillation model structure for application. The part of the ablation experiment is just the split and combination of each module, and more adequate experiments can be considered.

Response: Thank you for your suggestions. In the revision manuscript, we modified the model and supplemented the ablation experiments combined with different modules.

Case2: Two feature extractors E are used to obtain Eh. Is it redundant? Can it be replaced by a general feature extractor, with two different resolutions as input, so that E and D confront each other;

Response: Thank you for your question. Our idea is to enable feature extractor E to propose high-resolution features for low-resolution images. If the weight is shared, it is difficult to extract good high-resolution features for high-resolution images, so it is difficult to guide feature extractor E to extract high-resolution features for low-resolution images. Therefore, we set up two feature extractors Eh and E to ensure better quality of extracted high-resolution features.

Case3: The algorithm flow is not clear enough, the algorithm format is wrong, it should not be a screenshot;

Response: We revised it according to the algorithm’ style requirements, such as algorithm 1.

Case4: English translation has grammatical errors or has not been investigated, such as target detection.

Response: We have modified the English translation grammatical errors and the improper English expressions in this paper.

Case5: There is a lack of detailed process description below the paper frame diagram.

Response: Thank you very much for your suggestions. In the revision manuscript, we added a detailed process description under the framework chart of the paper.

Reviewer #4: 

Point 1: In the title, the short-phrase ‘Knowledge-guided and information-rich small target detection in foggy scenes 'could be rewritten making it be more understandable. Here knowledge and information are abstracted from the image because of fog or are used in a method/process?

Response: We modified the title as “Texture and semantic integrated small objects detection in foggy scenes”.

Point 2: In the abstract, the conclusion of ‘the existing small target detection algorithms ignore the texture information’ is not very precise. U can correct it with ‘some of the existing small target detection algorithms’. 

Response: There is something wrong with the content description. It should be “the existing small target detection algorithms ignore the texture information of tiny objects”. We modified it.

Point 3: The abstract should be rewritten carefully; then I could find how your experiment arranged logically.

Response: We have improved the abstract.

Point 4: Self-attention mechanism is applied in semantic information enhancement, and it is a very important part in this manuscript so I think it should be added in keywords. 

Response: “Self-attention mechanism” is a keyword, because the PLOS ONE's template does not reflect the keywords, it is not reflected in this manuscript.

Point 5: In lines 134-136, I recommend to add citations supporting the statement of self-attention. Because it is not the conclusion of the article itself, so it should be expressed clearly.

Response: There is a corresponding introduction in the cited reference [19]( Hu. H, Gu. J, Zhang. Z, Dai. J and Wei. Y. Relation Networks for Object Detection[C]. IEEE Conference on Computer Vision and Pattern Recognition (CVPR), Salt Lake City, USA, 2018:3588-3597.). Therefore, we do not specifically cite reference.

Point 6: According to the foggy images, have u used any method to defog these images? If u have had some experiments, please show the results in the manuscript. If not, u can defogging these images, and they can be input to ur proposed algorithm. If u don’t think it necessary, please tell me why.

Response: In the revision manuscript, we supplemented the ablation experiments combined with different modules in experiments section.

---

## [Decision Letter · Decision Letter 1]

9 May 2022

PONE-D-21-32064R1Texture and semantic integrated small objects detection in foggy scenesPLOS ONE

Dear Dr. Wang,

Thank you for submitting your manuscript to PLOS ONE. After careful consideration, we feel that it has merit but does not fully meet PLOS ONE’s publication criteria as it currently stands. Therefore, we invite you to submit a revised version of the manuscript that addresses the points raised during the review process.

The revised manuscript had been reviewed by 2 reviewers who reviewed the original version. Reviewer 2 was satisfied with the revisions made by authors and recommended accept. Reviewer 3  was also satisfied with revisions, however he had made some minor suggestions to further improve work and recommended minor revision. After consideration of comments of reviewers, my decision is "minor revision". Please incorporate the comments made by Reviewer 3.

We look forward to receiving your revised manuscript.

Kind regards,

Gulistan Raja

Academic Editor

PLOS ONE

Journal Requirements:

Reviewers' comments:

Reviewer's Responses to Questions

**Comments to the Author**

1. If the authors have adequately addressed your comments raised in a previous round of review and you feel that this manuscript is now acceptable for publication, you may indicate that here to bypass the “Comments to the Author” section, enter your conflict of interest statement in the “Confidential to Editor” section, and submit your "Accept" recommendation.

Reviewer #2: (No Response)

Reviewer #3: All comments have been addressed

2. Is the manuscript technically sound, and do the data support the conclusions?

Reviewer #2: (No Response)

Reviewer #3: Yes

3. Has the statistical analysis been performed appropriately and rigorously? 

Reviewer #2: (No Response)

Reviewer #3: Yes

4. Have the authors made all data underlying the findings in their manuscript fully available?

Reviewer #2: (No Response)

Reviewer #3: Yes

5. Is the manuscript presented in an intelligible fashion and written in standard English?

Reviewer #2: (No Response)

Reviewer #3: Yes

6. Review Comments to the Author

Reviewer #2: (No Response)

Reviewer #3: （1） The picture is not together with the corresponding title. The title of the table is in the wrong position and exceeds the limit of the body. The image resolution is too low and blurred to see the specific description of each module.

（2） In Figure 6, it is not recommended to use the line chart for the influence results of the super parameters, because the between each two parameters are irrational numbers, which are not easy to test, and do not necessarily increase or decrease linearly. If the line chart is used, it is obvious that the influence of the super parameters on map becomes a linear relationship, and the appearance is strange. It is recommended to use a histogram to show the impact of the tested parameters on map.

（3） There is no reference to others' methods in the method column of all tables.

（4） There are many data tables and module structure diagrams in the article, but there is no detection effect diagram of the experimental data set. In this paper, the small target detection in fog scene has achieved higher accuracy. It is suggested to give the target detection effect diagram of the model on the experimental data set and the target detection effect diagram of the baseline model for comparison.

（5） Table 2 shows the experimental results of different methods on the "coco" data set, in which there are AP, ap50, APS and other measurement indicators, which are not explained and explained in the article, which will cause misunderstanding.

7. PLOS authors have the option to publish the peer review history of their article (what does this mean?). If published, this will include your full peer review and any attached files.

Reviewer #2: No

Reviewer #3: **Yes: **Xing Wei

---

## [Author Response · Author response to Decision Letter 1]

7 Jun 2022

Thank you for allowing a resubmission of our manuscript, with an opportunity to address the reviewers’ comments.

Reviewer #3: 

Case1: The picture is not together with the corresponding title. The title of the table is in the wrong position and exceeds the limit of the body. The image resolution is too low and blurred to see the specific description of each module.

Response: When we submitted the manuscript of the paper, the editor made the following requirements, so it was modified.

Please note that PLOS does not allow figures to be included within the body of the manuscript. Before we can proceed, please remove the figures from the body of the manuscript, and upload them as separate figure files in accordance with our submission guidelines.

Case2: In Figure 6, it is not recommended to use the line chart for the influence results of the super parameters, because the between each two parameters are irrational numbers, which are not easy to test, and do not necessarily increase or decrease linearly. If the line chart is used, it is obvious that the influence of the super parameters on map becomes a linear relationship, and the appearance is strange. It is recommended to use a histogram to show the impact of the tested parameters on map.

Response: We change the line chart in Figure 6 into a histogram.

Case3: There is no reference to others' methods in the method column of all tables.

Response: We cited references for each method in Tables 1 and 2.

Case4: There are many data tables and module structure diagrams in the article, but there is no detection effect diagram of the experimental data set. In this paper, the small target detection in fog scene has achieved higher accuracy. It is suggested to give the target detection effect diagram of the model on the experimental data set and the target detection effect diagram of the baseline model for comparison.

Response: In the experiments section, we added visual analysis and visual comparison between yolov5 method and our method. It more intuitively proved the effectiveness of our method.

Case5: Table 2 shows the experimental results of different methods on the "coco" data set, in which there are AP, ap50, APS and other measurement indicators, which are not explained and explained in the article, which will cause misunderstanding.

Response: The meanings of AP and mAP are the same. In order not to cause ambiguity, AP has been modified to mAP in the paper, and then the evaluation index meanings of mAP50, mAP75, mAPS, mAPM and mAPL on the COCO data set are explained in the experimental evaluation index part.

---

## [Editor Report · Decision Letter 2]

9 Jun 2022

Texture and semantic integrated small objects detection in foggy scenes

PONE-D-21-32064R2

Dear Dr. Wang,

We’re pleased to inform you that your manuscript has been judged scientifically suitable for publication and will be formally accepted for publication once it meets all outstanding technical requirements.

Kind regards,

Gulistan Raja

Academic Editor

PLOS ONE
---

## [Editor Report · Acceptance letter]

10 Aug 2022

PONE-D-21-32064R2 

Texture and semantic integrated small objects detection in foggy scenes 

Dear Dr. Wang:

I'm pleased to inform you that your manuscript has been deemed suitable for publication in PLOS ONE. Congratulations! Your manuscript is now with our production department. 

Kind regards, 

on behalf of

Dr. Gulistan Raja 

Academic Editor

PLOS ONE